# NMR-Based Metabolomic Study of Purple Carrot Optimal Harvest Time for Utilization as a Source of Bioactive Compounds

**Fabio Sciubba** [1,2,*], **Alberta Tomassini** [1,2], **Giorgio Giorgi** [3], **Elisa Brasili** [4], **Gabriella Pasqua** [4], **Giorgio Capuani** [1,2], **Walter Aureli** [3] and **Alfredo Miccheli** [2,4]

1 Department of Chemistry, Sapienza University of Rome, P.le Aldo Moro 5, 00185 Rome, Italy; alberta.tomassini@uniroma1.it (A.T.); giorgio.capuani@uniroma1.it (G.C.)
2 NMR-Based Metabolomics Laboratory, Sapienza University of Rome, P.le Aldo Moro 5, 00185 Rome, Italy; alfredo.miccheli@uniroma1.it
3 R&D, Aureli Mario S.S. Agricola, Via Mario Aureli 7, 67050 Ortucchio (Aq), Italy; r.d@aurelimario.com (G.G.); produzione@aurelimario.com (W.A.)
4 Department of Environmental Biology, Sapienza University of Rome, P.le Aldo Moro 5, 00185 Rome, Italy; elisa.brasili@uniroma1.it (E.B.); gabriella.pasqua@uniroma1.it (G.P.)
* Correspondence: fabio.sciubba@uniroma1.it

**Abstract:** The carrot (*Daucus carota* L.), one of the most important vegetable crops in the world, is recognized as a source of different compounds with healthy properties. Due to their high content of anthocyanins, purple carrots have been used as a natural colorant source to face the increasing demand of consumers for non-synthetic products. However, the root developmental stage can greatly affect the phytochemical composition and, in this regard, the identification of chemical biomarkers for the optimal harvest time would be of paramount interest both from a nutritional point of view and for the agri-food industry. In the present work, the metabolic profiling of purple carrots was monitored over four months using high-resolution $^1$H NMR spectroscopy. Several metabolites were identified, and their quantitative variations allowed for the investigation of the carrot development processes. The metabolic profile analysis showed an increase in amino acid, NAD, and caffeic acid levels during carrot development. A more tardive harvest in December entailed an increase in levels of luteolin-7-O-glucoside, chlorogenic acid, falcarinol, and γ-aminobutyrate, and a decrease in carotenoids and ω-6 fatty acid. The results showed how the harvest time affects the composition in terms of flavonoids, phenols, and polyacetylenes, therefore, improving the bioactive compound content.

**Keywords:** *Daucus carota* ssp. *sativus* var. *atrorubens* Alef.; NMR-based metabolic profiling; carrot development; harvest time

## 1. Introduction

The carrot (*Daucus carota* L.) is a plant that belongs to the Apiaceae family and is one of the most important and least expensive vegetable crops in the world because of its fleshy, edible, colorful root. It is recognized worldwide as a source of different bioactive compounds with documented anti-inflammatory, antioxidant, antimicrobial, antiviral, and anticancer properties [1]. The edible portion of the carrot is its tuber-like root, which can be consumed either fresh or transformed into products such as juices. Carrots are important from a nutritional point of view since they provide phytochemicals such as carotenoids, anthocyanins, and other phenolic compounds without a significant amount of calories (43 kcal per 100 g of product). Historically, several varieties of carrots have been

cultivated, which differed mainly in color [2]. In particular, purple carrots (*Daucus carota* ssp. *sativus* var. *atrorubens*) contain about nine times more phenols and a considerable amount of anthocyanins in respect to carrots of other colors thus representing, beside their nutritional value, a good alternative source of non-synthetic food-derived colorants [3–5].

This is a very important aspect since foods high in flavonoids and antioxidants have been accepted by the scientific community as a good starting point for diets able to prevent the insurgence of several diseases [6].

However, the root growth and development process, together with other influencing factors (climate, cultivation site, agronomic practices), can greatly affect the phytochemical composition of the plant, as already observed for fruit ripening. Studies on the ripening of fruits have indeed shown that the harvest time can be chosen to take into account a specific or potential effect on consumer health in relation to the variation of the phytochemical components that increase or decrease during development [7,8].

Although the ripening process has been well documented for fresh fruits [9,10], the metabolic variations during root development have not been thoroughly studied, and a detailed investigation of the changes in the chemical composition is required.

In the case of carrots, the full development and consequential harvest time is a tricky subject, since it usually encompasses a period ranging from 75 to 130 days according to both cultivar and growth conditions, and the agronomic parameters chosen to assess the stage are essentially morphological, such as dimension, color intensity, crispness, and the shape of the apical region.

For the identification of metabolic biomarkers of maturation, a comprehensive NMR-based metabolomics approach was employed to follow the compositional changes during development.

In fact, nuclear magnetic resonance (NMR) spectroscopy can provide the identification and quantification of several molecular classes simultaneously, even in complex matrices, and has found increasing applications in metabolomics and food chemistry [11–13].

In the present study, high-resolution $^{1}$H NMR spectroscopy is employed to monitor the development of purple carrots, from September to November. To minimize the variability of the samples due to different types of soil and pedoclimatic conditions, a specific parcel of terrain was dedicated to the cultivation of the roots under investigation. Moreover, since it is well known that late-harvested carrots, especially of the purple variety, are characterized by a bitter taste [14], the development period was extended to the month of December for a better understanding of this process.

## 2. Materials and Methods

### 2.1. Plant Materials

Purple carrots (*Daucus carota* ssp. *sativus* var. *atrorubens*) were seeded and grown by Aureli Mario ss Agricola (Ortucchio, Aq, Italy). The location is situated at 41°52′ N latitude and 12°12′ E longitude with a mean altitude of 680 m above sea level.

Plants were sown in August 2018 and were grown in the same open field with the same environmental conditions and cultivation methods. Carrot roots were harvested in the months of September, October, November, and December, corresponding to the second, third, fourth, and fifth months of development, respectively.

### 2.2. Sample Preparation

Samples at each harvest time were ground and immediately stored at −80 °C. A total of 1 g of ground carrots was extracted following a modified Bligh-Dyer protocol [15]. Briefly, each aliquot was placed in a mortar, ground in liquid nitrogen, and added to a cold mixture composed of chloroform, methanol, and water in a 2:2:1 proportion. After an overnight incubation at 4 °C, the samples were centrifuged for 25 min at 4 °C with a rotation speed of 10,000 rpm on an Itettich Zentrifugen centrifuge (Germany). The upper hydrosoluble phase and the lower lipophilic phase were carefully separated

and dried under a gentle flow of nitrogen. The hydrophilic phase was resuspended in a mixture of $D_2O$/MeOD in a ratio of 2:1 containing 3-(trimethylsilyl)-propionic-2,2,3,3-$d_4$ acid sodium salt (TSP, 2 mM) as an internal chemical shift and concentration standard. The hydrophobic phase was resuspended in $CDCl_3$ with hexamethyldisiloxane (HMDS, 2 mM) as an internal standard. All solvents and standards were purchased from Sigma Aldrich (St. Louis, MO, USA).

*2.3. NMR Experiments*

All spectra were recorded at 298 K on a Bruker AVANCE III spectrometer (Bruker BioSpin, Karlsruhe, Germany), equipped with a Bruker multinuclear z-gradient inverse probe-head operating at the proton frequency of 400.13 MHz.

[1]H NMR spectra were acquired employing the *presat* pulse sequence for solvent suppression with 64 transients, a spectral width of 6009.13 Hz, and 65536 data points for an acquisition time of 5.5 s. The recycle delay was set to 6.55 s to achieve complete resonance relaxation between successive scans.

Bidimensional NMR experiments, [1]H–[1]H TOCSY, [1]H–[13]C HSQC, and [1]H–[13]C HMBC, were performed using the same experimental conditions previously reported [15]. Spectra were analyzed with ACD NMR manager software ver. 12 (ACD/Labs, Toronto, ON, Canada).

Only the molecules univocally identified were considered for the study, and their quantification was performed by integration of their NMR signals. Due to the overcrowding of [1]H NMR spectra, only those signals that did not overlap with other resonances were considered for integration. Quantities were expressed in µmol/g through comparison of the relative integrals with the reference concentration and normalized to the number of protons (TSP: 9 protons and HMDS: 18 protons) and to the fresh weight of carrots.

*2.4. Statistical Analysis*

Multivariate principal component analysis (PCA) [16] was performed on the data matrix with the Unscrambler ver. 10.5 software (Camo Software AS, Oslo, Norway). Data were mean-centered, since the variables with the largest response could dominate the PCA, and then autoscaled to equalize the importance of the variation of each variable.

Univariate one-way ANOVA was performed with SigmaPlot 14.0 software (Systat Software Inc., San Jose, CA, USA). The Shapiro-Wilk test was performed on each variable to assess data normality prior to one-way ANOVA. For the ANOVA-positive variables, a Holm-Sidak all pairwise multiple comparison test was applied to determine which categories were discriminated by these metabolites ($p < 0.05$).

## 3. Results

A total of 49 metabolites were identified and quantified from [1]H NMR spectra from both the hydroalcoholic and the chloroform extract. Resonance assignment was carried out on the basis of the signal chemical shift, multiplicity, TOCSY, HSQC, and HMBC correlations. For example, the assignment of chlorogenic acid was initially hypothesized due to the presence, in the [1]H hydroalcoholic spectra, of two doublets at 7.65 and 6.39 ppm, an ABX system at 7.19 (doublet), 7.12 (doublet of doublets), and 6.94 ppm (doublet) and a complex spin system at 5.33, 4.23, 3.88, 2.12, and 2.07 ppm, with the latter two resonances having an area double that of the others. The spin systems were identified due to the correlation among the proton resonances while the connection between the systems was determined by combining the information from HSCQ and HMBC experiments. In greater detail, HSQC provided the resonances of the carbons directly bonded to the observed protons and the HMBC showed the long-range correlations among the systems. Specifically, the HMBC correlation between the proton at 7.65 ppm with the carbon at 40.11 ppm indicated the presence of an ester bond between the caffeoyl and quinic moieties, thus allowing to univocally assign the chlorogenic acid. The particulars of the TOCSY, HSQC, and HMBC experiments concerning chlorogenic acid are reported in Figure 1, while a detail of the TOCSY experiment carried out on chloroform extract is reported in Figure 2 with evidence of the

resonances of falcarinol and carotenoids. The $^1$H chemical shifts, multiplicity, and the $^{13}$C chemical shifts of the identified molecules are reported in Supplementary Table S1.

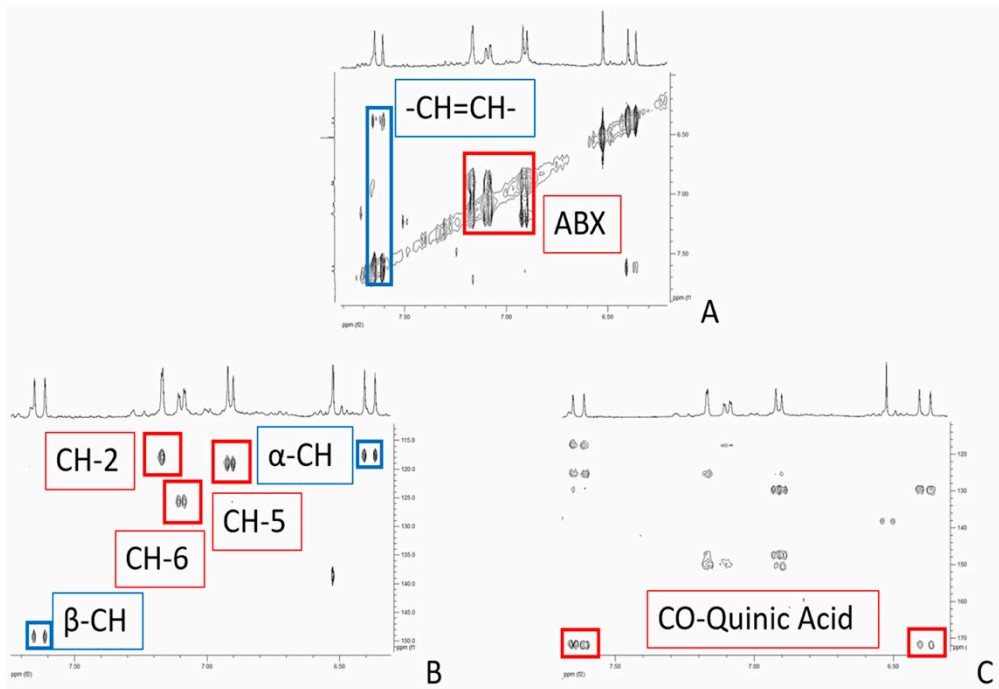

**Figure 1.** Details of TOCSY (**A**), HSCQ (**B**), and HMBC (**C**) spectra of the hydroalcoholic extract with evidence of the diagnostic resonances of chlorogenic acid.

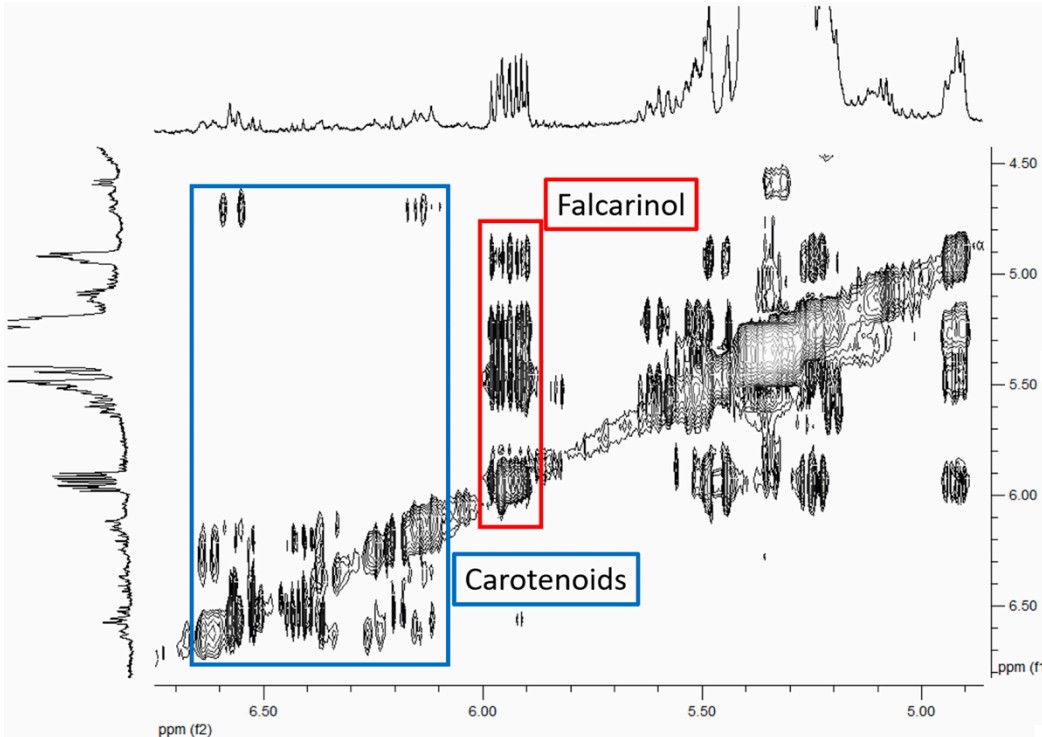

**Figure 2.** TOCSY spectrum of the chloroform extract with evidence of the diagnostic resonances of falcarinol and carotenoids.

The chosen experimental parameters did not allow to quantify the anthocyanins. In fact, while purple carrots are known to contain anthocyanins acylated with caffeic and other hydroxycinnamic acids [17], the pH of the samples (about pH 6.5) does not allow for their detection due to the short $T_2$ relaxation time of the diagnostic resonance of the hydrogen in position four of their aglycone moiety [18]. Nevertheless, these experimental conditions were chosen since they allow for the quantifying of a greater number of molecular species due to a reduced superimposition of their diagnostic resonances.

To determine the optimum harvest time of purple carrots, a PCA analysis was carried out on the whole data matrix, providing a model whose first six components explained 80% of the overall variance (Figure S1). In this model, a spontaneous grouping of the samples according to their development time was observed, but the analysis of the loadings could not differentiate the processes occurring during the root development from the ones active in late-harvested carrots. As such, two more PCA analyses were performed, one taking into account the carrots harvested during the development months (September, October, and November), and the other was carried out on the profiles of carrots during the late period growth (November and December).

The first PC analysis considered carrots from the second to the fourth month of development and provided a model whose first six components explained 80% of the overall variance (Figure 3), with the first component (PC1) accounting for 40% and the second one (PC2) for 12% of the overall variance.

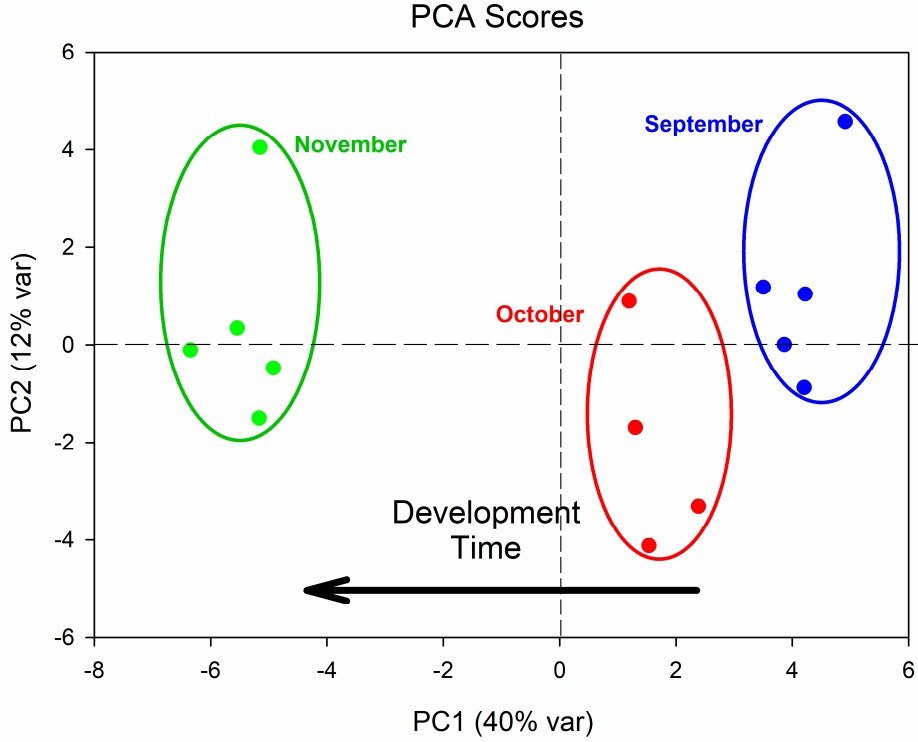

**Figure 3.** PCA scores of the purple carrot. Carrots harvested in September are evidenced in blue, October in red, and November in green.

Even in this model, a spontaneous grouping of the samples according to their harvest time was evident, and it was possible to determine that PC1 captures variance related to development time.

The normalized loading plot of the first component (Figure 4) shows which variables are important along with PC1 factor. The Pearson table for critical values for correlation was consulted, and variables with normalized loading values greater than 0.7 and lower than −0.7 were considered significant for the model ($p < 0.05$). Monoacylglycerols, 1,2-propanediol, and fumaric acid are the metabolites positively correlated with PC1 meaning that they are more abundant in carrots harvested in September. The levels of tryptophan, valine, isoleucine, leucine, threonine, glutamine, asparagine, inosine triphosphate,

NAD, quinic acid, aspartic acid, acetic acid, histidine, phenylalanine, alanine, and caffeic acid are higher in carrots harvested at later months.

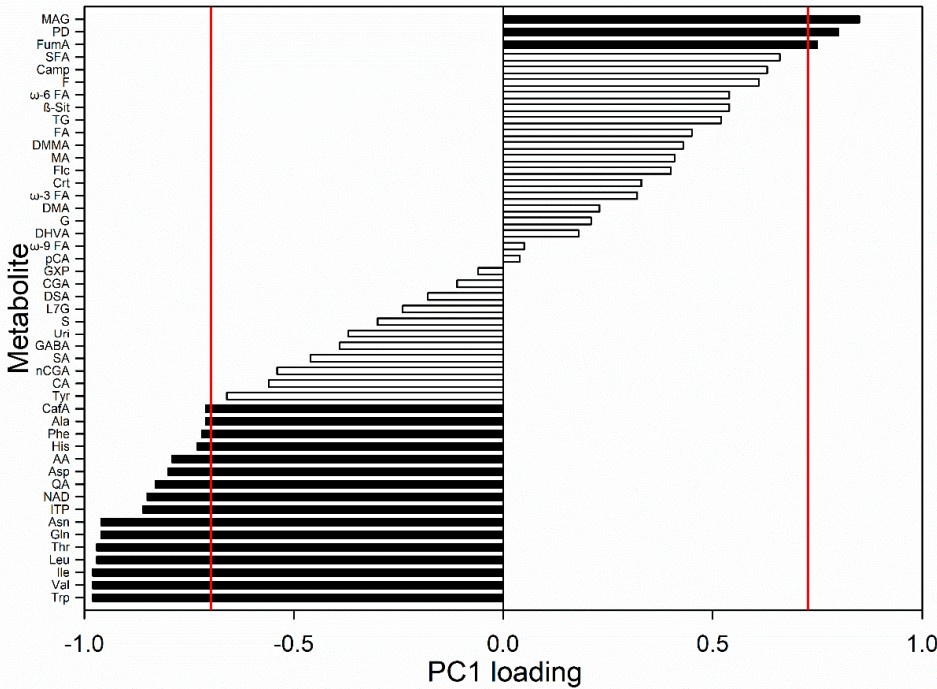

**Figure 4.** PC1 normalized loadings of purple carrot harvested in September, October, and November. Variables with $p < 0.05$ are presented in black and molecule abbreviations are reported in the Abbreviations.

In Table 1, only the concentrations of the metabolites identified as important for the separation of carrot harvest time according to PCA analysis are reported.

**Table 1.** Comparison among carrots harvested in September, October, and November for molecules resulting to be significant by PCA analysis, and one-way-ANOVA was employed to assess statistical differences.

| Metabolite | Amount (µmol/g) | | |
|---|---|---|---|
| | September | October | November |
| Leucine | 0.26 ± 0.03 | 0.63 ± 0.06 § * | 1.93 ± 0.14 † ** |
| Isoleucine | 0.08 ± 0.02 | 0.25 ± 0.03 § * | 0.93 ± 0.05, † ** |
| Valine | 0.12 ± 0.01 | 0.37 ± 0.03 § * | 1.15 ± 0.07 † ** |
| 1,2-propanediol | 0.82 ± 0.08 | 0.38 ± 0.03 § ** | 0.25 ± 0.03 |
| Threonine | 0.07 ± 0.01 | 0.14 ± 0.01 § * | 0.31 ± 0.02 † * |
| Alanine | 0.38 ± 0.05 | 1.11 ± 0.19 § ** | 1.19 ± 0.11 |
| Quinic acid | 1.04 ± 0.11 | 1.01 ± 0.05 | 1.52 ± 0.05 |
| Acetic acid | 0.24 ± 0.02 | 0.26 ± 0.02 | 0.65 ± 0.12 † * |
| Glutamine | 2.45 ± 0.55 | 5.06 ± 0.91 | 11.65 ± 0.64 † * |
| Aspartic acid | 0.39 ± 0.07 | 0.46 ± 0.13 | 0.94 ± 0.11 † * |
| Asparagine | 0.61 ± 0.09 | 1.13 ± 0.16 | 2.08 ± 0.11 |
| ITP | 0.021 ± 0.002 | 0.015 ± 0.003 | 0.06 ± 0.01 † * |
| Caffeic acid | 0.26 ± 0.01 | 0.18 ± 0.04 | 0.49 ± 0.07 † ** |
| Fumaric acid | 0.44 ± 0.02 | 0.40 ± 0.03 | 0.30 ± 0.03 |
| Phenylalanine | 0.021 ± 0.003 | 0.07 ± 0.01 | 0.09 ± 0.01 |
| Tryptophan | 0.04 ± 0.01 | 0.11 ± 0.03 § ** | 0.38 ± 0.03 † ** |
| Histidine | 0.21 ± 0.02 | 0.13 ± 0.02 § ** | 0.39 ± 0.04 † ** |
| NAD | 0.037 ± 0.002 | 0.037 ± 0.004 | 0.049 ± 0.001 |
| Monoacylglycerol | 0.75 ± 0.08 | 0.30 ± 0.02 § ** | 0.09 ± 0.01 |

§: significant differences between September and October and †: between October and November; * $p < 0.05$; ** $p < 0.01$.

The results show a linear increase of leucine, isoleucine, valine, threonine, and tryptophan levels from September to November. On the contrary, the other amino acids have a different evolution: histidine decreases from September to October and increases in November; alanine increases in October and remains stable in November; glutamine and aspartic acid levels increase in November. Even acetic acid and caffeic acid increase in November, while ITP decreases in November; 1,2-propanediol and monoacylglycerols decrease in October and then remain stable in November.

To better understand the metabolic changes occurring in carrots harvested after the fourth month, PCA was conducted on the samples of roots collected in November and December (Figure 5).

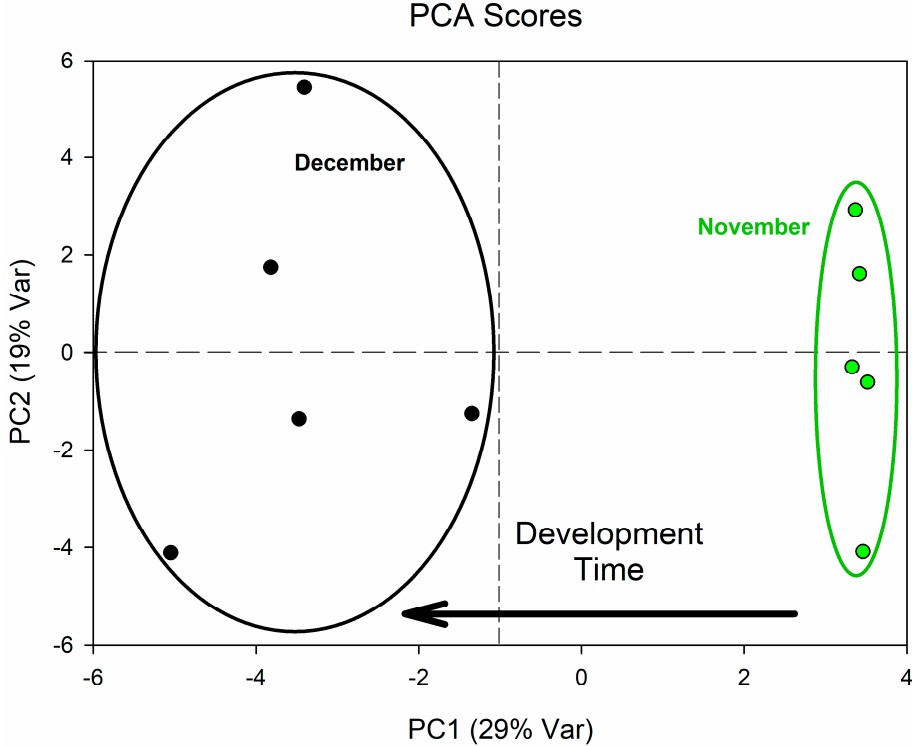

**Figure 5.** PCA scores of the purple carrot. Carrots harvested in November are in green and December in black.

This analysis provided a six-component model whose PC1 accounted for 29% of the overall variance and whose PC2 accounted for 19%. Even in this model, it was possible to identify a time axis that was inversely proportional to the PC1 values since all samples harvested in November had a positive PC1 value, while samples harvested in December showed negative PC1 values.

From the analysis of the loading plot (Figure 6), the variables that were positively correlated ($p < 0.05$) with PC1, and thus that decreased from November to December were leucine, carotenoids, aspartic acid, 1,2-propanediol, and $\omega$-6 polyunsaturated fatty acids. Metabolites that increased during late harvest ($p < 0.05$) were alanine, falcarinol, GABA, NAD, luteolin 7-O-glucoside, tyrosine, chlorogenic acid, and fumaric acid.

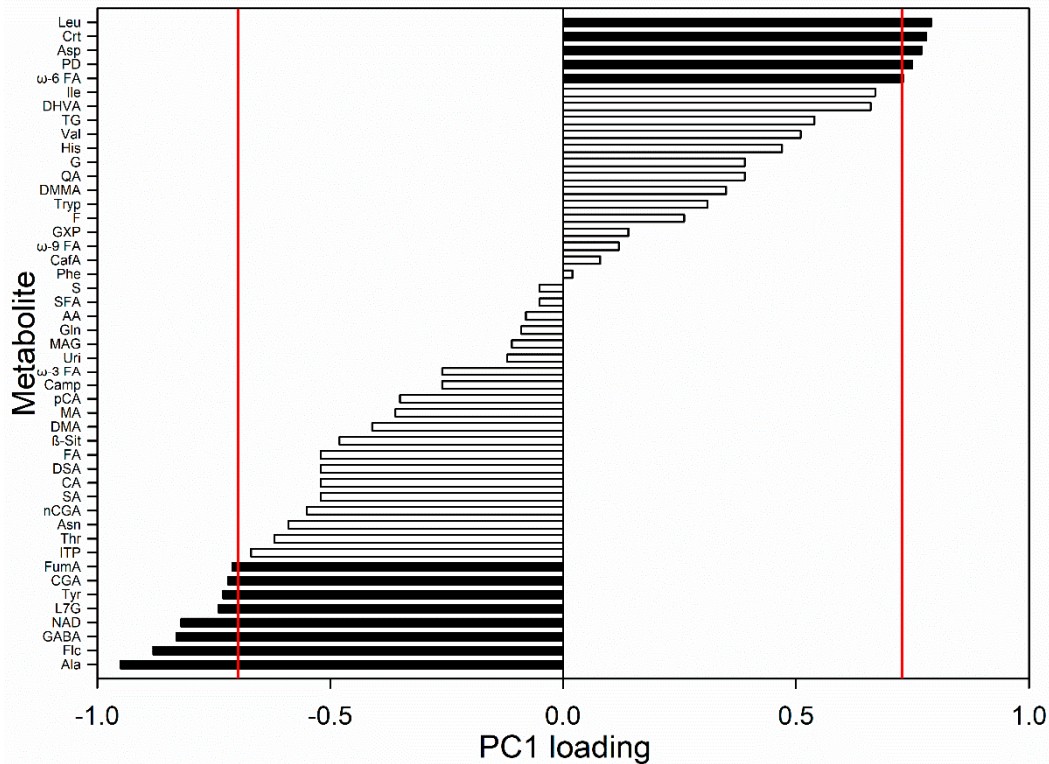

**Figure 6.** PC1 normalized loadings of purple carrot harvested in November and December. Variables with $p < 0.05$ are presented in black and molecule abbreviations are reported in the Abbreviations.

The univariate analysis conducted on these molecules and reported in Table 2 confirmed what was observed in the multivariate analysis: the carrots harvested in December had greater amounts of alanine, GABA, falcarinol, luteolin 7-O-glucoside, chlorogenic acid, NAD, tyrosine, and fumaric acid while having lower levels of leucine, carotenoids, 1,2 propanediol, aspartic acid, and ω-6 polyunsaturated fatty acids.

**Table 2.** Comparison of the carrots harvested in November and December for the molecules resulting to be significant by PCA analysis, and one-way-ANOVA was employed to assess statistical differences.

| Metabolite | Amount (μmol/g) | |
|---|---|---|
| | November | December |
| Leucine | 1.93 ± 0.14 | 1.18 ± 0.13 ** |
| 1,2-Propanediol | 0.25 ± 0.03 | 0.13 ± 0.02 ** |
| Alanine | 1.19 ± 0.11 | 2.61 ± 0.12 ** |
| GABA | 1.27 ± 0.13 | 3.05 ± 0.33 ** |
| Aspartic acid | 0.94 ± 0.11 | 0.51 ± 0.05 * |
| Luteolin 7-O-Glucoside | 0.83 ± 0.08 | 1.18 ± 0.12 * |
| Chlorogenic acid | 1.14 ± 0.15 | 2.09 ± 0.41 * |
| Fumaric acid | 0.30 ± 0.03 | 0.65 ± 0.18 * |
| Tyrosine | 1.08 ± 0.08 | 1.68 ± 0.25 * |
| NAD | 0.054 ± 0.002 | 0.11 ± 0.02 * |
| ω-6 unsaturated fatty acids | 2.62 ± 0.24 | 1.42 ± 0.26 * |
| Falcarinol | 0.26 ± 0.04 | 0.54 ± 0.03 ** |
| Carotenoids | 0.009 ± 0.002 | 0.0031 ± 0.0004 ** |

$* \ p < 0.05; ** \ p < 0.01.$



## 4. Discussion

The investigation of the ripening process is of paramount interest not only from a scientific point of view but also for the agri-farm industries facing the increasing consumer awareness about the healthiness and safety of their food [19].

In this regard, metabolic profiling has been applied to study the development of fleshy fruits such as tomatoes, grapes, strawberries, and peaches among others [20–23].

Quite surprisingly, there are considerably fewer studies on the metabolic variations in roots, especially edible ones, although both development and ripening can have a significant influence on their phytochemical composition, as demonstrated by investigations on targeted compounds such as carotenoids, auxins, and plant hormones or on different cultivars [24–28].

Furthermore, the chemical composition of the raw material is affected by factors such as different cultivars, geographical origin, growing sites, and soil, seasonal, and climatic differences [15,29]. This source of intrinsic variability is further increased by the choice of the harvest time, which should be standardized when evaluating the potentiality of a food in terms of either nutrient content and health benefits, or different utilization of the raw material or derived products such as, for example, natural colorants or source of bioactive compounds.

PCA of the purple carrot composition as a function of time showed a clear clustering of the samples along PC1 with a separation of the carrots harvested in September, October, and November (Figure 3).

The model was affected by a sharp increase in the free amino acids stored in the roots during maturation. This increase suggests that different metabolic processes are involved in root development compared to fruit ripening since an overall decrease in amino acid levels coupled with the induction of catabolic transcripts has been observed during fruit maturation [30]. Furthermore, while fruits accumulate sugars during ripening, sucrose, glucose, and fructose levels did not change significantly in carrot roots during the three sampling times.

It is known that the regulation of amino acid catabolism varies among plant species, tissues of the same species, and their developmental stages [30]. In addition, amino acids are involved in many physiological processes such as energy production, redox state regulation, resistance to abiotic and biotic stresses, as well as plant growth and development, and are precursors of many different secondary products such as phenylpropanoids [31]. Amino acid content has also been linked to the sensing of nitrogen status, alterations in exudate composition, and inhibition of the level of root tissues [32,33]. Therefore, the observed high increase in free amino acid content in carrot roots during maturation could be the result of both physiological processes during growth and seasonally linked environmental changing conditions. In this regard, the sharp increase in the levels of the amino acids leucine, isoleucine, and tryptophan from October to November could suggest that these compounds act as chemical biomarkers of full maturation for purple carrots (Table 1).

The observed increase in the precursors of anthocyanins, especially aromatic amino acids and phenylpropanoids during ripening is in agreement with the increase in anthocyanins observed both in different cultivars of purple carrots reported in the literature and in relation to the harvest time [34,35]. The concomitant increase during ripening of compounds possessing co-pigmentation enhancing properties, like the uncolored caffeic acid and NAD, could act as color stabilizers further increasing the value of carrot by-products in relation to the harvest time.

Chlorogenic and caffeic acids are secondary phenolic metabolites produced by tea, green roasted beans, coffee, berry fruits, cocoa, citrus fruits, apples, and pears and have been found at higher contents in purple carrots in respect to differently colored varieties, with the first one representing 72.5% of the total phenolic compounds [36,37].

The observed two-fold increase of caffeic and chlorogenic acids, as well as the increase of about 30% of luteolin 7-O-glucoside content from November to December, makes the latter month suitable as the optimal harvest time due to the particularly high content of these bioactive compounds with

recognized beneficial activities for health like antioxidant, anti-inflammatory, anti-obesity, antidiabetic, and antihypertensive activities [38–40].

Finally, to investigate the optimal harvest time, we analyzed the purple carrots after an additional month of cultivation, as a bitter taste in this cultivar is known to develop due to overripening. It is also known that this taste is usually associated with the presence of bioactive compounds [41]. This change in their sensorial properties makes late-harvested carrots unsuitable for consumer consumption, but they could be employed as a source of bioactive compounds.

Indeed, our results (Table 2) showed an increase in GABA, chlorogenic acid, luteolin-7-O-glucoside, and polyacetylenes content and a decrease in branched amino acids and polyunsaturated $\omega$-6 fatty acid. These metabolic trends could be induced by an abrupt change in environmental conditions. In fact, we must remember that the fifth month of cultivation corresponds to the month of December, which is characterized by both autumnal abundant rain and an abrupt drop in temperature in the Fucino area (www.meteoam.it). Due to the rain, the bacterial and fungal charge in the soil could increase [42], and the plant could respond by producing phenylpropanoids, which have antibacterial properties, and polyacetylenes, which have antifungal activity [43,44]. The increase in polyacetylenes is correlated with the decrease in polyunsaturated $\omega$-6 fatty acid levels in carrots since they are the precursors of both falcarinol and falcarindiol [45]. Both molecule classes are also known to confer a bitter taste to foods and, as such, are the origin of this flavor in late-harvested carrots. Furthermore, it has been demonstrated that waterlogging reduces oxygen availability in the soil, causing severe stress to the roots that respond through the activation of the GABA shunt pathway [46–49]. The activation of this pathway in purple carrot roots is supported by the observed increase in GABA, alanine, and formic acid and the decrease in aspartic acid levels from November to December. In agreement with our hypothesis, the trend of these molecules has been reported to be a consequence of different hypoxic treatments in almost all the studied species [50]. More specifically, it has been proposed that alanine synthesis could represent a defense mechanism that decreases the excess pyruvate produced in anaerobic conditions and pyruvate metabolization to avoid the increase in lactic acid and ethanol in hypoxia [51].

## 5. Conclusions

This study identified several biomarkers that can be employed to identify the optimal harvest time of purple carrots in order to differentiate between roots to be employed as food and others to be used as a source of bioactive compounds. In particular, since amino acids increase during development and decrease in the late phases, their regular quantification is a useful tool to choose the desired carrot application given that their decrease indicates a sharp change in the metabolic processes occurring in the roots, which was associated with an increase in stored bioactive compounds like flavonoids, phenylpropanoids, and polyacetylenes.

**Supplementary Materials:** The following are available online at http://www.mdpi.com/2076-3417/10/23/8493/s1, Table S1: Metabolites identified in the $^1$H NMR spectrum of the aqueous and chloroform extracts of purple carrots; Figure S1: Score plot of PCA analysis carried out on purple carrot samples.

**Author Contributions:** Conceptualization, A.M. and W.A.; methodology, A.T., F.S., and E.B.; software, F.S. and E.B.; validation, G.C.; formal analysis, F.S.; investigation, G.G. and F.S.; data curation, G.G.; writing—original draft preparation, F.S. and A.T.; writing—review and editing, A.M. and A.T.; visualization, F.S. and G.G.; supervision, A.M., G.P., and W.A.; project administration, A.M.; funding acquisition, A.M. and W.A. All authors have read and agreed to the published version of the manuscript.

**Funding:** The present work has been carried out under the project "INNOPAQ" funded by POR-FESR Abruzzo 2014-2020, CUP number: C73D18000320007.

**Conflicts of Interest:** The authors declare no conflict of interest.

**Abbreviations**

2,3-Dihydroxyvaleric acid (DHVA), Acetic acid (AA), Alanine (Ala), Asparagine (Asn), Aspartic acid (Asp), β-Sitosterol (β-ST), Caffeic acid (CafA), Campsterol (Camp), Carotenoids (Crt), Chlorogenic acid (CGA), Choline (Chn), Citric acid (CA), p-Coumaric Acid (pCA), Dimethyl Amine (DMA), Dimethyl Malonic acid (DMMA), Dimethyl Succinc acid (DMSA), Falcarinol (Flc), Formic acid (FA), Fructose (F), Fumaric acid (FumA), γ-aminobutyric acid (GABA), Glucose (G), Glutamine (Gln), Guanosine n-phosphate (GXP), Histidine (His), Inosine Triphosphate (ITP), Isoleucine (Ile), Lactic acid (LA), Leucine (Leu), Luteolin 7-glucoside (L7G), Malic acid (MA), Monoacylglycerol (MAG), Monounsaturated ω-9 fatty acid (ω-9 FA), Neochlorogenic acid (nCGA), Nicotinamide adenine dinucleotide (NAD), Saturated fatty acid (SFA), Phenylalanine (Phe), Polyunsaturated ω-3 fatty acid (ω-3 FA), Polyunsaturated ω-6 fatty acid (ω-6 FA), Propane-1,2-Diol (PD), Quinic acid (QA), Succinic acid (SA), Sucrose (S), Threonine (Thr), Triglyceride (TG), Tryptophan (Trp), Tyrosine (Tyr), Uridine (Uri), Valine (Val).

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
