# Peer review of "NMR-Based Metabolomic Study of Purple Carrot Optimal Harvest Time for Utilization as a Source of Bioactive Compounds"

_applsci, doi:10.3390/app10238493_

Round 1

Reviewer 1 Report

In the present work the metabolic profiling of purple carrots was carried out by the authors for a period of four months using high-resolution 1H NMR spectroscopy. Several metabolites were succesfully identified, and authors demosntarted their quantitative variations allowed the carrot development
processes. The work is interesting anc could be accepted in the current form.

Author Response

The authors kindly thanks the reviewer for the comments.

Reviewer 2 Report

In this study, the authors use NMR profiling and PCA analysis to monitor the metabolic change of the compounds contained in purple carrot, identifying the several biomarkers associated with ripening. Thus, they claims that this study has developed a way to find the optimal harvest time of purple carrots. The research is clearly presented and easy to follow. However, I have a significant concern about the generality of the results.

The authors used the crops cultivated in a specific area in Italy. Thus, the current results and the biomarkers will only be applicable to this area. When the temperature, humidity, soil, and altitude etc. are different, this should affect the growth and metabolic profiles in the roots of the carrots. In this regard, it might be better to modify the title of this paper to include the information on the cultivated place because the current title gives the readership the impression that the present findings can be applied to all the carrots grown in different conditions. And the authors should describe the comparison of their results with similar studies on carrots cultivated in different climatic conditions.

The following are minor points:
Fig.1 the characters in the graphs are too small to be recognized.

In Fig. 5, the distribution of the data points in the PCA scores is much larger for December than for November. What does this suggest?

In Table 1, there is no data corresponding to GABA. Is it below the detection limit on September and October?

Author Response

The authors used the crops cultivated in a specific area in Italy. Thus, the current results and the biomarkers will only be applicable to this area. When the temperature, humidity, soil, and altitude etc. are different, this should affect the growth and metabolic profiles in the roots of the carrots. In this regard, it might be better to modify the title of this paper to include the information on the cultivated place because the current title gives the readership the impression that the present findings can be applied to all the carrots grown in different conditions. And the authors should describe the comparison of their results with similar studies on carrots cultivated in different climatic conditions.

This is the first study concerning the development of purple carrots, and as such no other studies are reported in literature, making it impossible to compare the results of our observations with the ones of other research groups. Nonetheless, we discussed our data comparing them with other studies which correlated, for example, the amino acid content with the temperature, as well as the secondary metabolite content with humidity. While the reviewer is correct while stating that several pedoclimatic factors can affect the metabolic profile of plant organs (our research group did several studies in that regard), it is also true that the root responses to these conditions are general and even common among several plant species.

The following are minor points:
Fig.1 the characters in the graphs are too small to be recognized.

Figure 1 has been improved according to the reviewer’s request.

In Fig. 5, the distribution of the data points in the PCA scores is much larger for December than for November. What does this suggest?

It is possible that carrots, during the late stage of maturation, could start a lignification process that could affect the uniformity of sampling. Such process did not occur in carrots harvested at earlier months. Increases in carrot root diameter are primarily due to continuous differentiation of the vascular cambium, a plant tissue located between the xylem and phloem [Wang GL, Jia XL, Xu ZS, Wang F, Xiong AS (2015) Sequencing, assembly, annotation, and gene expression: novel insights into the hormonal control of carrot root development revealed by a high-throughput transcriptome. Mol Gen Genomics 290:1379–1391]. The vascular cambium produces secondary xylem toward the carrot root interior and secondary phloem toward the exterior; this process is controlled by several plant hormones leading to a higher lignin content in the roots. The initial phase of this process could explain the higher observed variance of December samples.

In Table 1, there is no data corresponding to GABA. Is it below the detection limit on September and October?

In Table 1, as well as Table 2, only the metabolites which have significant PCA loading values are reported. GABA was not considered because its changes were not significant in the PCA of the months September-October-November. The Table captions have been modified to better clarify this point.

Reviewer 3 Report

General comments

The authors describe the NMR metabolomics analysis of carrot roots sampled at 4 different time points during development. The timepoints are chosen to based on a desire to determine an "optimal" harvest time for different applications, namely consumption as a food product, and natural dye extraction.

Please consider making the data publicly available e.g. via Metabolights (https://www.ebi.ac.uk/metabolights/). It would be an excellent example dataset for training purposes.

In general the study appears carefully considered, and is well described. I include some minor comments below.

Introduction

Line 43

Reference [Arscott & Tanumihardjo, 2010]  should be numbered and included in the list of references

Plant Materials

Were the carrots harvested from random locations within the plot for each time point? How to you know that the soil is consistent across the plot e.g there isn’t a big lump of clay or something that interferes with growth of the carrots that you harvested in a particular month?

I suppose it is not possible to separate the effects of weather from the development of the root in your study. Mentioning the impact of weather, and if there were any significant changes in weather during the experiment here may be of value, especially since you bring it up later during the discussion.

Statistical analysis

Please define the acronym "PCA" and consider providing a reference.

1-way ANOVA was performed on each variable. Was a multiple test correction such as FDR or Bonferroni applied? It would be advisable to do so as with 49 features there is a high probability of false positives just due to the number of tests being applied.

Shapiro-Wilks test was used to assess normality prior to one-way ANOVA; I assume all features were accepted as normal? If they were not, was a different test statistic used e.g. Kruskal-Wallis? For biological studies a log transform is often required. Was this explored?

Figure S1

What do the ellipses represent? 95% T2 ellipses would be ellipsoids in 3 dimensional space, so it is unclear what these 2d ellipses represent.

Line 177

'inversely proportional' implies a ratio. I would rephrase this to something like "PC1 captures variance related to development time".

Line 179

What is a "Fisher table for covariance significance"? How have you applied this to the loadings? Please clarify.

Line 192

Replace "resulting" with "identified as"

Figure 4

Please indicate where we can find the meaning of the abbreviations used here. EDIT: I found them at the end of the document; please refer the reader to it in the text.

Figure 4 caption

Rephrase "In black the variables with p<0.05 are evidenced" with something like "Variables with p<0.05 are presented in black". Likewise Fig 6.

Figure 5

Do you have comments on why there appears to be greater variance in the December harvest?

Table 1

How do you have p values for individual months? PCA analysis was applied to all three months at the same time, so the p-value should be for the metabolite, not for the concentration within a month. If you have subsequently applied 1-way ANOVA to the metabolites, please make this clear in the text.

Discussion

Line 258

Figure 5 doesn’t include Sep Oct and Nov, only Nov and Dec.

Line 263

"levels remain almost constant". I don’t think you have evidence of this, and it's not presented anywhere since you only include significant metabolites in the tables. I suggest changing the wording to "[fructose etc] levels did not change significantly throughout our experiment".

Line 286-288

The wording used here is hard for me to understand. Rephrase to something like "The observed 2 fold increase in caffeic acid, the 2 fold increase in chlorogenic acid and an increase in luteolon of approximately 30% makes these metabolites strong candidates for the selection of harvest time, because…"

Author Response

Introduction

Line 43

Reference [Arscott & Tanumihardjo, 2010] should be numbered and included in the list of references

The manuscript was modified according to the reviewer’s request.

Plant Materials

Were the carrots harvested from random locations within the plot for each time point? How to you know that the soil is consistent across the plot e.g there isn’t a big lump of clay or something that interferes with growth of the carrots that you harvested in a particular month?

The soil parcels destined for this study were carefully analyzed before seeding to avoid this specific issue. Moreover, the carrots examined at the same harvest time possessed similar sizes (i.e. overall length and weight), thus suggesting that they developed at the same rate.

I suppose it is not possible to separate the effects of weather from the development of the root in your study. Mentioning the impact of weather, and if there were any significant changes in weather during the experiment here may be of value, especially since you bring it up later during the discussion.

According to the climatic data from region Abruzzo (https://www.regione.abruzzo.it/archivio-dati-climatici-settimanali), the main changes occurred between November and December, with a sharp decrease of the temperatures as well as an increase of rainfall.

Statistical analysis

Please define the acronym "PCA" and consider providing a reference.

The manuscript was modified according to the reviewer’s request.

1-way ANOVA was performed on each variable. Was a multiple test correction such as FDR or Bonferroni applied? It would be advisable to do so as with 49 features there is a high probability of false positives just due to the number of tests being applied.

The 1-way ANOVA was performed using Holm-Sidak test as multiple comparison option, but according to the reviewer’s suggestion, a Bonferroni test was also applied, but no differences between the results of the two tests were observed.

Shapiro-Wilks test was used to assess normality prior to one-way ANOVA; I assume all features were accepted as normal? If they were not, was a different test statistic used e.g. Kruskal-Wallis? For biological studies a log transform is often required. Was this explored?

All molecules concentrations were normal and, given their comparable concentration range, a log transform was not necessary.

Figure S1

What do the ellipses represent? 95% T2 ellipses would be ellipsoids in 3 dimensional space, so it is unclear what these 2d ellipses represent.

The ellipses are only a graphical tool to better visualize the sample grouping.

Line 177

'inversely proportional' implies a ratio. I would rephrase this to something like "PC1 captures variance related to development time".

The sentence was modified according to the reviewer’s request.

Line 179

What is a "Fisher table for covariance significance"? How have you applied this to the loadings? Please clarify.

The sentence was modified and clarified according to the reviewer’s request.

Line 192

Replace "resulting" with "identified as"

The sentence was modified according to the reviewer’s request.

Figure 4

Please indicate where we can find the meaning of the abbreviations used here. EDIT: I found them at the end of the document; please refer the reader to it in the text.

The captions were modified according to the reviewer’s suggestions.

Figure 4 caption

Rephrase "In black the variables with p<0.05 are evidenced" with something like "Variables with p<0.05 are presented in black". Likewise Fig 6.

The captions of Figure 4 and Figure 6 were modified according to the reviewer’s suggestions.

Figure 5

Do you have comments on why there appears to be greater variance in the December harvest?

It is possible that carrots, during the late stage of maturation, could start a lignification process that could affect the uniformity of sampling. Such process did not occur in carrots harvested at earlier months. Increases in carrot root diameter are primarily due to continuous differentiation of the vascular cambium, a plant tissue located between the xylem and phloem [Wang GL, Jia XL, Xu ZS, Wang F, Xiong AS (2015) Sequencing, assembly, annotation, and gene expression: novel insights into the hormonal control of carrot root development revealed by a high-throughput transcriptome. Mol Gen Genomics 290:1379–1391]. The vascular cambium produces secondary xylem toward the carrot root interior and secondary phloem toward the exterior; this process is controlled by several plant hormones leading to a higher lignin content in the roots. The initial phase of this process could explain the higher observed variance of December samples.

Table 1

How do you have p values for individual months? PCA analysis was applied to all three months at the same time, so the p-value should be for the metabolite, not for the concentration within a month. If you have subsequently applied 1-way ANOVA to the metabolites, please make this clear in the text.

The captions of Table 1 and Table 2 were changed according to the reviewer’s suggestions.

Discussion

Line 258

Figure 5 doesn’t include Sep Oct and Nov, only Nov and Dec.

The reference to the figure was corrected.

Line 263

"levels remain almost constant". I don’t think you have evidence of this, and it's not presented anywhere since you only include significant metabolites in the tables. I suggest changing the wording to "[fructose etc] levels did not change significantly throughout our experiment".

The sentence was corrected according to the reviewer’s indication.

Line 286-288

The wording used here is hard for me to understand. Rephrase to something like "The observed 2-fold increase in caffeic acid, the 2-fold increase in chlorogenic acid and an increase in luteolon of approximately 30% makes these metabolites strong candidates for the selection of harvest time, because…"

The sentence was rewritten according to the reviewer’s suggestion.

Round 2

Reviewer 2 Report

The authors now responded the review's comments one-by-one. The manuscript is now suitable for publication.